# Practical continuous-variable quantum key distribution with composable security

Nitin Jain ◉[1] ✉, Hou-Man Chin ◉[1,2], Hossein Mani ◉[1], Cosmo Lupo ◉[3,4],
Dino Solar Nikolic[1], Arne Kordts[1], Stefano Pirandola ◉[5],
Thomas Brochmann Pedersen[6], Matthias Kolb[7], Bernhard Ömer[7],
Christoph Pacher ◉[7], Tobias Gehring ◉[1] ✉ & Ulrik L. Andersen ◉[1] ✉

A quantum key distribution (QKD) system must fulfill the requirement of universal composability to ensure that any cryptographic application (using the QKD system) is also secure. Furthermore, the theoretical proof responsible for security analysis and key generation should cater to the number $N$ of the distributed quantum states being finite in practice. Continuous-variable (CV) QKD based on coherent states, despite being a suitable candidate for integration in the telecom infrastructure, has so far been unable to demonstrate composability as existing proofs require a rather large $N$ for successful key generation. Here we report a Gaussian-modulated coherent state CVQKD system that is able to overcome these challenges and can generate composable keys secure against collective attacks with $N \approx 2 \times 10^8$ coherent states. With this advance, possible due to improvements to the security proof and a fast, yet low-noise and highly stable system operation, CVQKD implementations take a significant step towards their discrete-variable counterparts in practicality, performance, and security.

Quantum key distribution (QKD) is the only known cryptographic solution for distributing secret keys to users across a public communication channel while being able to detect the presence of an eavesdropper[1,2]. In an ideal case, legitimate QKD users (Alice and Bob) encrypt their messages with the secret keys and exchange them with the assurance that the eavesdropper (Eve) cannot break the confidentiality of the encrypted messages.

In one of the most well-known flavors of QKD, the quantum information is coded in continuous variables[2–5], such as the amplitude and phase quadratures of the optical field, described by an annihilation operator $\hat{a}$. Alice encodes random bits, e.g., by modulating the optical signal field to obtain a coherent state that follows the relation $\hat{a}_{\text{sig}}|\alpha\rangle = \alpha_{\text{sig}}|\alpha\rangle$, with the real [imaginary] part of the complex value $\alpha_{\text{sig}}$ equal to the amplitude [phase] quadrature.

Bob decodes this information using coherent detection, facilitated by a so-called local oscillator (LO), that yields a quantity $\propto \beta_{\text{LO}}\hat{b}_{\text{sig}}^{\dagger} + \beta_{\text{LO}}^{*}\hat{b}_{\text{sig}}$ for an incoming field operator $\hat{b}_{\text{sig}}$ and with $|\beta_{\text{LO}}|^2$ as the LO intensity.

Figure 1 shows these steps of quantum state preparation, transmission (on a quantum channel) and measurement, which Alice and Bob perform in the beginning of the continuous-variable (CV)QKD protocol. The quantum stage is followed by classical data processing steps and a security analysis, performed in accordance with a mathematical "security" proof, to obtain a key of a certain length. For this purpose, Alice and Bob use an authenticated channel on which Eve cannot modify the communicated messages but can learn their content. Once the classical stage concludes, Alice and Bob use their secret keys to encrypt their messages, and the resulting ciphertexts are

[1]Center for Macroscopic Quantum States (bigQ), Department of Physics, Technical University of Denmark, 2800 Kongens Lyngby, Denmark. [2]Department of Photonics, Technical University of Denmark, 2800 Kongens Lyngby, Denmark. [3]Department of Physics and Astronomy, University of Sheffield, Sheffield S3 7RH, UK. [4]Dipartimento Interateneo di Fisica, Politecnico di Bari, 70126 Bari, Italy. [5]Department of Computer Science, University of York, York YO10 5GH, UK. [6]Cryptomathic A/S, Aaboulevarden 22, 8000 Aarhus, Denmark. [7]Center for Digital Safety & Security, AIT Austrian Institute of Technology GmbH, 1210 Vienna, Austria. ✉e-mail: nitinj@iitbombay.org; tobias.gehring@fysik.dtu.dk; ulrik.andersen@fysik.dtu.dk

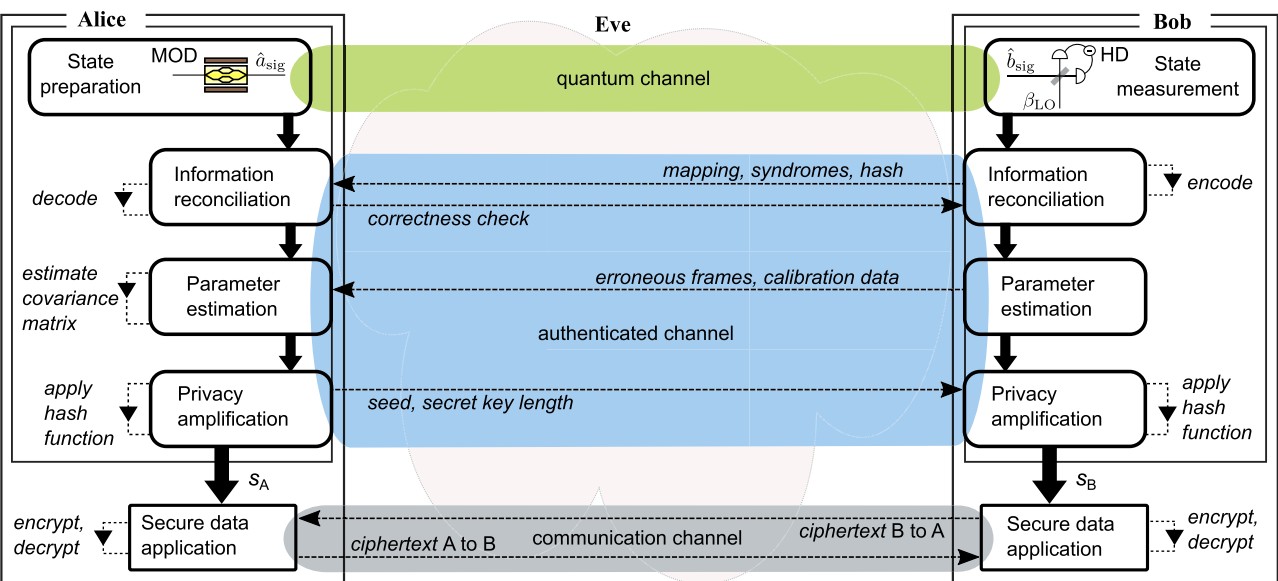

**Fig. 1 | Composability in continuous-variable quantum key distribution (CVQKD) with coherent states.** Alice and Bob obtain quantum correlations over the quantum channel by means of modulation (MOD) and local oscillator (LO) aided homodyne/heterodyne detection (HD) to prepare and measure, respectively, optical coherent states. After going through the remaining steps of the protocol that involve the authenticated channel, they obtain correlated bitstreams $s_A$ and $s_B$, respectively. Certain criteria associated with correctness, robustness, and secrecy of the protocol must be satisfied, for the application to assure composable security[7,10]. For instance, $\epsilon$-correctness implies that Alice and Bob possess the same symmetric key $s (= s_A = s_B)$ except with a probability $\epsilon_{cor}$ that bounds the probability of them having non-identical keys ($\Pr[s_A \neq s_B] \leq \epsilon_{cor}$). This key can be used for encrypting a message and decrypting the corresponding ciphertext across the communication channel. Dashed lines with arrows indicate classical communication across the channel and local operations. Eve is assumed to control all the channels. Further details of our CVQKD protocol implementation are presented in later sections of this article.

exchanged using a communication channel, e.g., a telephone line, and decrypted.

Amongst the many physical considerations included in the security proof, Eve's actions on the channels (particularly her interaction with the transmitted quantum states) are classified in the form of individual, collective, or general attacks, in increasing order of power and generality[1,2]. For instance, a security proof catering to a *collective attack* permits Eve to store the result of her interactions with the quantum states in a quantum memory, and later perform a collective measurement. Also, the fact that Alice and Bob cannot avail an infinite number of quantum states in practice adversely affects the key length but such *finite-size corrections* are essential for the security assurance. Another related property of a secret key is *composability*[6], which allows specifying the security requirements for combining different cryptographic applications in a unified and systematic way. In the context of practical QKD, composability is of utmost importance because the secret keys obtained from a protocol are used in other applications, e.g. data encryption[7]. A secret key not proven to be composable is thus practically useless.

Composable security in CVQKD was first proven[8] and experimentally demonstrated[9] using two-mode squeezed states, but the achievable communication distance was rather limited since the employed entropic uncertainty relation is not tight. Composable proofs for CVQKD systems using coherent states and dual quadrature detection, first proposed in 2015[10], have been progressively improved[11–15]. Some of these proofs even provide security against general attacks, but all promise keys at distances much longer than in ref. 8 apart from the advantage of dealing with coherent states, which are much easier to generate than squeezed states.

Nonetheless, the strongest proof[16] that actual coherent-state CVQKD implementations, e.g., refs. 17–21, have used so far unfortunately does not include composable definitions. An experimental demonstration of composability in CVQKD has thus remained elusive, and this is due to a combination of the strict security bounds (because of a complex parameter estimation routine), the large number of required quantum state transmissions (to keep the finite-size terms sufficiently low), and the stringent requirements on the tolerable excess noise.

In this article, we demonstrate a CVQKD setup of low complexity that is capable of generating composable keys secure against collective attacks. We achieve this by deriving a method for establishing confidence intervals compatible with collective attacks, which allows us to work on smaller (and thus more practical) block sizes than originally required[10]. Alice produces coherent states by encoding Gaussian information in frequency (side-)bands shifted away from the optical carrier[22] by means of a single electro-optical in-phase and quadrature (IQ) modulator. Bob decodes this information using *real* LO-assisted radio frequency (RF) heterodyning, implemented with a single balanced detector, followed by digital signal processing (DSP)[23]. By performing a careful analysis to either eradicate or avoid various spurious noise components, and by implementing a machine learning framework for phase compensation[24], we are able to keep the excess noise below the null key length threshold. After taking finite-size effects as well as confidence intervals from various system calibrations into account, we achieve a positive composable key length with merely $N \approx 2 \times 10^8$ coherent states (also referred to as 'quantum symbols' from hereon) transmitted over a 20 km long fiber-optic channel. With $N = 10^9$, we obtain > 41 Mbits worth of key material that is composably secure against collective attacks, assuming worst-case confidence intervals.

## Results
### Composably secure key
A DSP routine at the end of the quantum stage yields the digital quantum symbols discretized with $d$ bits per quadrature. This stream is divided into $M$ frames for information reconciliation (IR), after which we perform parameter estimation (PE) and privacy amplification (PA); as visualized in Fig. 1. We derive the secret key bound for reverse reconciliation, i.e., Alice correcting her data according to Bob's quantum symbols $\bar{Y}$.

The (composable) secret key length $s_n$ for $n$ coherent state transmissions is calculated using tools from refs. [10],[15] as well as results presented in the following. The key length is bounded per the leftover hash lemma in terms of the smooth min-entropy $H_{\min}^{\epsilon_s}$ of $\bar{Y}$ conditioned on the quantum state of the eavesdropper $E$ with $\epsilon_s$ as the smoothing parameter[25]. From this we subtract the information reconciliation leakage $\text{leak}_{IR}(n, \epsilon_{IR})$ and obtain,

$$s_n^{\epsilon_h + \epsilon_s + \epsilon_{IR}} \geq H_{\min}^{\epsilon_s}(\bar{Y}|E)_{\rho^n} - \text{leak}_{IR}(n, \epsilon_{IR}) + 2\log_2(\sqrt{2}\epsilon_h). \quad (1)$$

The security parameter $\epsilon_h$ characterizes the hashing function and $\epsilon_{IR}$ describes the failure probability of the correctness test after IR.

The probability $p'$ that IR succeeds in a frame is related to the frame error rate (FER) by $p' = 1 - \text{FER}$. All frames in which IR failed are discarded from the raw key stream, and this step thereby projects the original tensor product state $\rho^n \equiv \rho^{\otimes n}$ into a non i.i.d. state $\tau^n$. To take this into account, one replaces the smooth min-entropy term in Eq. (1) with the expression[15]:

$$H_{\min}^{\epsilon_s}(\bar{Y}|E)_{\tau^{n'}} \geq H_{\min}^{\frac{p'}{3}\epsilon_s^2}(\bar{Y}|E)_{\rho^{\otimes n'}} + \log_2\left(p' - \frac{p'}{3}\epsilon_s^2\right), \quad (2)$$

where $n' = np'$ is the number of quantum symbols remaining after error correction.

The asymptotic equipartition property (AEP) bounds the conditional min-entropy in the following way,

$$H_{\min}^{\delta}(\bar{Y}|E)_{\rho^{\otimes n'}} \geq n'H(\bar{Y}|E)_\rho - \sqrt{n'}\Delta_{\text{AEP}}(\delta, d), \quad (3)$$

where

$$\Delta_{\text{AEP}}(\delta, d) \leq 4(d+1)\sqrt{\log_2(2/\delta^2)}, \quad (4)$$

is an improved penalty (proof provided in the "Methods" section) in comparison to ref. [10],[15] and the conditional von-Neumann entropy $H(\bar{Y}|E)_\rho$ from Eq. (3) is given by

$$H(\bar{Y}|E)_\rho = H(\bar{Y}) - I(\bar{Y}; E)_\rho. \quad (5)$$

We estimate the Shannon entropy $H(\bar{Y})$ directly from the data (up to a probability $\leq \epsilon_{\text{ent}}$, further details in the "Methods" section). The second term is Eve's Holevo bound with respect to $\bar{Y}$ that satisfies,

$$I(\bar{Y}; E)_\rho \leq I(Y; E)_\rho \leq I(Y; E)_{\rho_G},$$

where $Y$ is the continuous version of $\bar{Y}$ and $I(Y; E)_{\rho_G}$ is the Holevo information obtained by using the extremality property of Gaussian attacks.

The Holevo information is estimated by evaluating the covariance matrix using worst-case estimates for its entries based on confidence intervals. We improved the confidence intervals of ref. [10] by exploiting the properties of the Beta distribution. Let $\hat{x}, \hat{y}, \hat{z}$ be the estimators for the variance of the transmitted ensemble of coherent states, the received variance and the co-variance, respectively. The true values $y$ and $z$ are bound by

$$y \leq (1 + \delta_{\text{Var}}(n, \epsilon_{\text{PE}}/2))\hat{y}, \quad (6)$$

$$z \geq \left(1 - 2\delta_{\text{Cov}}(n, \epsilon_{\text{PE}}/2)\frac{\sqrt{\hat{x}\hat{y}}}{\hat{z}}\right)\hat{z} \quad (7)$$

with $\epsilon_{\text{PE}}$ denoting the failure probability of parameter estimation, and

$$\delta_{\text{Var}}(n, \epsilon) = a'(\epsilon/6)\left(1 + \frac{120}{\epsilon}e^{-\frac{n}{16}}\right) - 1,$$

$$\delta_{\text{Cov}}(n, \epsilon) = \frac{1}{2}\left[\frac{a'(\epsilon/6) - b'(\epsilon/6)}{2} + a'\left(\frac{\epsilon^2}{324}\right) - b'\left(\frac{\epsilon^2}{324}\right)\right]$$

being the confidence intervals (derived in Supplementary Note 1). In the above equations,

$$a'(\epsilon) = 2\left[1 - \text{invcdf}_{\text{Beta}(n/2, n/2)}(\epsilon)\right],$$
$$b'(\epsilon) = 2\,\text{invcdf}_{\text{Beta}(n/2, n/2)}(\epsilon),$$

where "invcdf" is the inverse cumulative distribution function. As detailed in section "Discussion", the (length of the) secret key we eventually obtain in our experiment requires an order of magnitude lower $N$ due to these confidence intervals.

Finally, we remark here on a technical limitation arising due to the digitization of Alice's and Bob's data. In practice, it is impossible to implement a *true* Gaussian protocol because the Gaussian distribution is both unbounded and continuous, while realistic devices have a finite range and bit resolution[14],[26]. In our work, we consider a range of 7 standard deviations and use $d = 6$ bits, leading to a constellation with $2^{2d} = 4096$ coherent states. Per recent results[27],[28], this should suffice to minimise the impact of digitization on the security of the protocol. For keeping the analysis simple, we however assume perfect Gaussian modulation.

## Experimental implementation

Figure 2 shows the schematic of our setup, consisting of a transmitter and a receiver connected together by a 20 km long standard single mode fiber spool, which formed the quantum channel. We performed optical single sideband modulation with carrier suppression (OSSB-CS) using an optical source (Tx laser) from NKT Photonics, and an IQ modulator plus automatic bias controller (IQmod+ABC) from ixBlue. An arbitrary waveform generator (AWG) was connected to the RF ports to modulate the sidebands. The coherent states were produced in a $B = 100$ MHz wide frequency sideband, shifted away from the optical carrier[22],[29]. Random numbers drawn from a Gaussian distribution obtained by transforming the uniform distribution of a vacuum-fluctuation based quantum random number generator (QRNG) with a security parameter $\epsilon_{\text{qrng}} = 2 \times 10^{-6}$ formed the complex amplitudes of these coherent states[30]. To this broadband 'quantum data' signal, centered at $f_u = 200$ MHz, we multiplexed in frequency a 'pilot tone' at $f_p = 25$ MHz for sharing a phase reference with the receiver[23],[31–33]. The left inset of Fig. 2 shows the complex spectra of the RF modulation signal.

After propagating through the quantum channel, the signal field's polarization was manually tuned to match the polarization of the real local oscillator (RLO) for heterodyning[31–33]. The Rx laser that supplied the RLO was free-running with respect to the Tx laser and detuned in frequency by ~320 MHz, giving rise to a beat signal, as labeled in the solid-red spectral trace in the right inset of Fig. 2. The quantum data band and pilot tone generated by the AWG are also labeled. Due to finite OSSB[29], a suppressed pilot tone is also visible; the corresponding suppressed quantum band was however outside the receiver bandwidth (we used a low pass filter with a cutoff frequency around 360 MHz at the output of the homemade heterodyne detector[30]). As shown, the Tx and Rx had their clocks synchronized, and the Tx provided a trigger for data acquisition in Rx[34],[35].

Separately, we also measured the vacuum noise (Tx laser off, Rx laser on) and the electronic noise of the detector (both Tx and Rx lasers off), depicted by the dotted-blue and dashed-green traces, respectively, in the right inset of Fig. 2. The clearance of the vacuum noise over the electronic noise is >15 dB over the entire quantum data band.

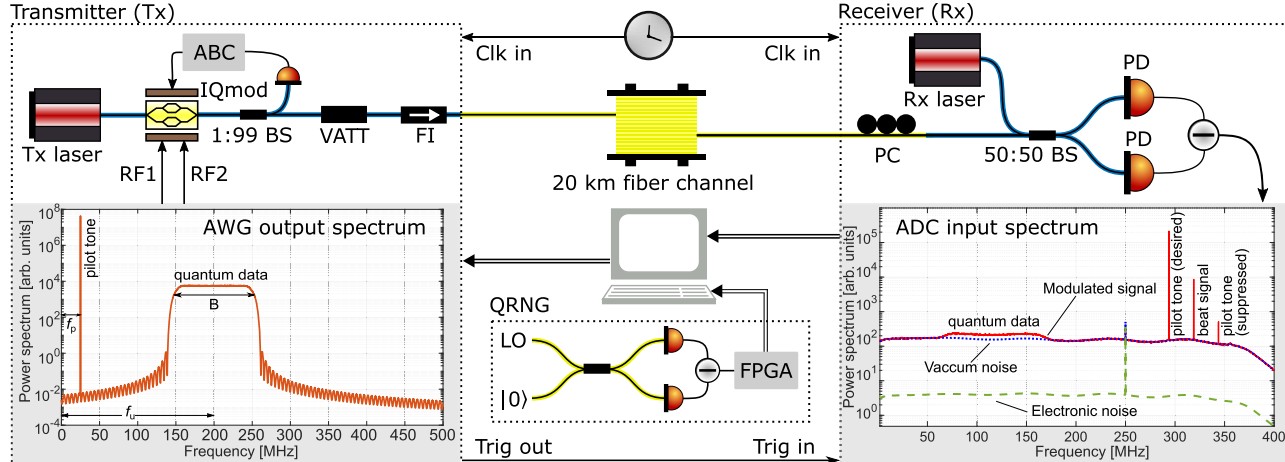

**Fig. 2 | Schematic of the experiment.** The transmitter (Tx) and receiver (Rx) were built from polarization maintaining fiber components. The transmitter comprised a 1550 nm continuous-wave laser (Tx laser), an in-phase and quadrature electro-optic modulator (IQmod) with automatic bias controller (ABC) for carrier suppression and single sideband modulation, and a variable attenuator (VATT) and Faraday isolator (FI). An arbitrary waveform generator (AWG) with 16 bit resolution and sampling rate of 1 GSps supplied waveforms RF1 and RF2 for driving IQmod. A quantum random number generator (QRNG) delivered Gaussian-distributed symbols for discrete Gaussian modulation of coherent states. The receiver comprised a laser (Rx laser; same type as Tx laser), a polarization controller (PC) to tune the incoming signal field's polarization, a symmetric beam splitter followed by a homemade balanced detector for RF heterodyning. The detector's output was sampled by a 16 bit analog-to-digital converter (ADC) at 1 GSps. BS: beam splitter, PD: photo detector. Left inset: Power spectrum of the complex waveform RF1 + ι RF2 driving the IQmod. Right inset: Power spectra of the receiver from 3 different measurements described in section "Experimental implementation". The noise peak at 250 MHz is an interleaving spur of the ADC.

## Noise analysis & calibration

A careful choice of the parameters defining the pilot tone and the quantum data band and their locations with respect to the beat signal is crucial in minimizing the excess noise. A strong pilot tone enables more accurate phase reference but at the expense of higher leakage in the quantum band and an increased number of spurious tones. The latter may arise as a result of frequency mixing of the (desired) pilot tone with e.g., the beat signal or the suppressed pilot tone. As can be observed in the right inset of Fig. 2, we avoided spurious noise peaks resulting from sum- or difference-frequency generation of the various discrete components (in the solid-red trace) from landing inside the wide quantum data band.

In CVQKD, it is well known that Alice needs to optimize the modulation strength of the coherent state alphabet at the input of the quantum channel to maximize the secret key length. For this, we connected the Tx and Rx directly, i.e., without the quantum channel, and performed heterodyne measurements to calibrate the *mean photon number* $\mu$ of the coherent states' ensemble. The AWG electronic gain and the variable attenuator (VATT) provided a fine-grained knob to control the modulation strength.

Since we conducted our experiment in the non-paranoid scenario[1,26], i.e., we trusted some parts of the overall loss and excess noise by assuming them to be beyond Eve's control, some extra measurements and calibrations for the estimation of trusted parameters become necessary. More specifically, we decomposed the total transmittance and excess noise into respective trusted and untrusted components. In Supplementary Note 4, we present the details of the calibration of the receiver efficiency (trusted transmittance) $\tau = 0.69$ and trusted noise from the detector $\xi_t = 25.71 \times 10^{-3}$ photon number unit (PNU). Let us remark here that in our work, we express the noise and other variance-like quantities, e.g., the modulation strength, in PNU as opposed to the traditional shot noise unit (SNU). The former is independent of quadratures and facilitates a comparison with discrete-variable (DV) QKD systems[36], highlighted using $\mu$ in Table 1. A simple factor of 2 relates these units: 1 photon number unit (PNU) corresponds to a variance of 2 shot noise units (SNU). Finally, note that we recorded a total of $10^{10}$ ADC samples for each of the calibration measurements, and all the acquired data was stored on a hard drive for offline processing.

## Protocol operation

After setting $\mu = 1.45$ PNU, we connected the Tx and Rx using the 20 km channel, optimized the signal polarization, and then collected heterodyne data using the same Gaussian distributed random numbers as mentioned above. Offline DSP[24] provided the symbols that formed the raw key. The preparation and measurement was performed with a total of $10^9$ complex symbols. After discarding some symbols due to a synchronization delay, Alice and Bob had a total of $N_{IR} = 9.88 \times 10^8$ correlated symbols at the beginning of the classical phase of the protocol, the implementation of which we describe below. Note that we assumed the existence of an authenticated channel for these steps.

1.  IR was based on a multi-dimensional scheme[37] using multi-edge-type low-density-parity-check error correcting codes[38]. As shown in Fig. 1, Bob sent the mapping and the syndromes, together with the hashes computed using a randomly chosen Toeplitz function, to Alice, who performed correctness confirmation and communicated it to Bob. We obtained a reconciliation efficiency $\beta = 94.3\%$ and FER = 12.1% for the experimental data. In Supplementary Note 5, we provide further details about the operating regime and the performance of these codes. Due to the non-zero FER, Alice and Bob had $N_{PA} = 8.69 \times 10^8$ complex symbols for distilling the secret key via PA.

2.  During PE, Alice estimated the entropy of the corrected symbols, and together with the symbols from the erroneous frames, i.e., frames that could not be reconciled successfully (and were publicly announced by Bob), Alice evaluated the covariance matrix. This was followed by evaluating the channel parameters using the receiver calibration data, performing the 'parameter estimation test' (refer Theorem 2 in ref. [10]), and getting a bound on Eve's Holevo information. Subtracting $\xi_t$ from the total excess noise of 30.9 mPNU yielded the mean untrusted noise $\xi_u = 30.9 - 25.7 = 5.2$ mPNU, while dividing the total transmittance of 0.25 by $\tau$ gives us the mean untrusted transmittance $\eta = 0.25/0.69 = 0.36$.

3.  Alice calculated a secret key length $l = 41378264$ bits in the worst-case scenario by substituting in Eq. (1) the security parameters $\epsilon_h = \epsilon_{ent} = \epsilon_{cal} = \epsilon_s = \epsilon_{PE} = 10^{-10}$ and $\epsilon_{IR} = 10^{-12}$, and $n = 2N_{PA}$ (factor of 2 owing to data from both I and Q quadratures). As shown in Fig. 1, this length was communicated together with a seed to Bob to

**Table 1 | Comparison of notable parameters from prepare-and-measure QKD experiments conducted in the last decade with similar physical channels, as indicated by the column showing loss and length**

| | Protocol implementation keywords | Loss/ Length [dB/km] | $\mu$ [PNU] | $\xi_u$ [PNU] | $\xi_Q$ [%] | B [MHz] | N($\times 10^9$) [symbols] | SKF [bits/symbol] |
|---|---|---|---|---|---|---|---|---|
| Huang et al. (ref. 33) | CV, Gaussian, Homodyne via RLO | 5.0/25.0 | 2.00 | 0.025* | | 100 | 0.02 | 0.0010* |
| Xu et al. (ref. 40) | DV, phase coding, Avalanche diode | 4.5/20.0 | 0.37 | | 2.73 | 5 | 7.84 | 0.0001* |
| Islam et al. (ref. 41) | DV, time-bin coding, Superconducting | 4.0/20.0 | 0.45 | | 5.49 | 2500 | 62.5 | 0.0105 |
| Wang et al. (ref. 20) | CV, Gaussian, Heterodyne via RLO | 5.0/25.0 | 1.62 | 0.011 | | 100 | 0.01 | 0.0185 |
| Zhang et al. (ref. 21) | CV, Gaussian, Homodyne via TLO | 4.4/27.3 | 7.19 | 0.002 | | 5 | 100 | 0.0560 |
| Current work | CV, Gaussian, Heterodyne via RLO | 4.6/20.3 | 1.45 | 0.005 | | 100 | 1 | 0.0471 |

Values with a superscript * may be somewhat inaccurate as they were inferred from a graph. $\mu$, mean photon number of the quantum state alphabet; $\xi_u$, untrusted noise (referred to channel output); $\xi_Q$, quantum bit error rate; B, repetition rate in pulsed or quantum data bandwidth in CW implementations; N, number of transmitted quantum data symbols or pulses in the experiment; secret key fraction (SKF), secret key length in bits divided by N. It is possible to parametrize $\xi_u$ and $\xi_Q$ by the same quantity, namely the mean number of noise photons from the channel, in CV and DV systems, respectively[36]. Also, assuming symmetry between the quadratures, 1 photon number unit (PNU) corresponds to a variance of 2 shot noise units (SNU).
R/TLO: real/transmitted LO.

select a random Toeplitz hash function. Alice and Bob then employed the high-speed and large-scale PA scheme[39] to generate the final secret key $s(=s_A=s_B)$. Note that the final security parameter $e^{(coll)}$ quantifying composable security against collective attacks is a linear summation of the various epsilons mentioned before; see Supplementary Note 2 for an exact expression.

## Discussion

Using the equations presented in section "Composably secure key", we can calculate the composably secure key length for a certain number $n$ of the quantum symbols. We partitioned $N = 10^9$ in 25 blocks, estimated the key length considering the total number $N_k$ of symbols accumulated from the first $k$ blocks, for $k \in \{1, 2, ..., 25\}$. Dividing this length by $N_k$ yields the composable secret key fraction (SKF) in bits/symbols. If we neglect the time taken by data acquisition, DSP, and the classical steps of the protocol, i.e., only consider the time taken to modulate $N = N_k$ coherent states at the transmitter (at a rate $B = 100$ MSymbols/ s), we can construct a hypothetical time axis to show the evolution of the CVQKD system.

Figure 3a depicts such a time evolution of the SKF after proper consideration to the finite-size corrections due to the average and worst-case (black and red data points, respectively) values of the underlying parameters. Similarly, Fig. 3b shows the experimentally measured untrusted noise $\xi_u$ (lower squares) together with the worst-case estimator (upper dashes) calculated using $N_k$ in the security analysis. To obtain a positive key length, the worst-case estimator must be below the maximum tolerable noise—null key fraction threshold—shown by the dashed line, and this occurs at $N/B \approx 2.0$ s.

Note that in reality, the DSP and classical data processing consume a significantly long time: In fact, we store the data from the state preparation and measurement stages on disks and perform these steps offline. The plots in Fig. 3 therefore may be understood to be depicting the time evolution of the SKF and the untrusted noise *if* the entire protocol operation was in real time.

Referring to Fig. 3a, the solid-red and dashed-black traces simulate the SKF in the worst-case and average scenarios, respectively, while the dotted-orange trace shows the asymptotic SKF value (with FER taken into account) obtainable with the given channel parameters. Per projections based on the simulation, the worst-case composable SKF should be within 5% of the asymptotic value for $N \approx 10^{11}$ complex symbols.

From a theoretical perspective, the reason for being able to generate a positive composable key length with a relatively small number of coherent states ($N \approx 2 \times 10^8$) can mainly be attributed to the improvement in confidence intervals during PE; refer Eqs. (6) and (7). Figure 3c and d quantitatively compare the scaling factor in the RHS of these equations, respectively, as a function of $N$ for three different distributions. The estimators $\hat{x}, \hat{y}, \hat{z}$ for this purpose are the actual values obtained in our experiment and we used an $\epsilon_{PE} = 10^{-10}$. The difference between the confidence intervals used in ref. 10 (suitably modified here for a fair comparison) with those derived here, based on the Beta distribution, is quite evident at lower values of $N$, as visualized by comparing the dashed-blue trace with the solid-red one.

Since the untrusted noise has a quadratic dependence on the covariance in contrast to variance where the dependence is linear, a method that tightens the confidence intervals for the covariance can be expected to have a large impact on the final composable SKF. In fact, if we had used the confidence intervals of Ref. 10, our implementation would not have produced any composable key until $N = 10^9$, at which the worst-case SKF would have been $6.04 \times 10^{-4}$, i.e., almost two orders of magnitude lower than what we have achieved here (single blue data point in bottom-right corner of Fig. 3a).

On the practical front, a reasonably large transmission rate $B = 100$ MSymbols/s of the coherent states together with the careful analysis and reduction of untrusted noise (refer section "Noise analysis & calibration" for more details) enables an overall fast, yet low-noise and highly stable system operation, critical in quickly distributing raw correlations of high quality and keeping the finite-size corrections minimal. Table 1 provides a comparison of results from our proof-of-concept experiment with three other Gaussian-modulated CVQKD experiments[20,21,33] that provide security against collective attacks but do not include composable security definitions. Table 1 also lists two[40,41] of (multiple) DVQKD experiments that have been able to prove composable security against general attacks in a realistic finite size regime—the holy grail for any QKD system. In the "Methods" section, we discuss the challenges for our CVQKD implementation in achieving this security criterion.

In conclusion, our results have demonstrated composability and protection against collective attacks while ensuring robustness against finite-size effects in a coherent-state CVQKD protocol, operating in laboratory conditions, over a 20 km long quantum channel. With an order of magnitude larger $N$ and half the current value of $\xi_u$, we expect to obtain a non-zero length of the composable key while tolerating channel losses around 8 dB, i.e., distances up to ~ 40 km (assuming an

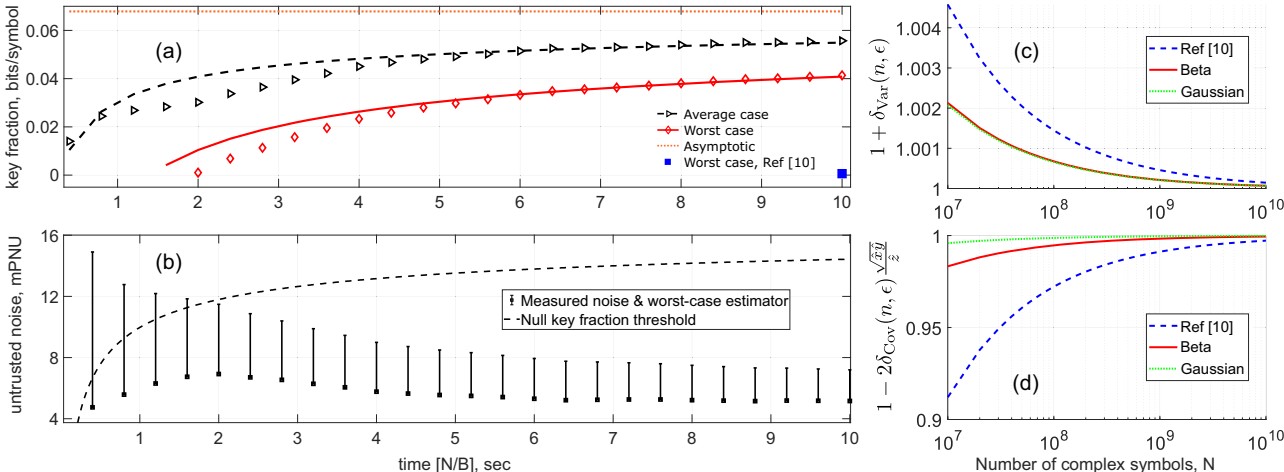

**Fig. 3 | Composable SKF results. a** Pseudo-temporal evolution of the composable SKF with the time parameter calculated as the ratio of the cumulative number $N$ of complex symbols available for the classical steps of the protocol and the rate $B = 100$ MHz at which these symbols are modulated. **b** Variation of untrusted noise $\xi_u$ measured in the experiment (lower point) and its worst-case estimator (upper point), and the noise threshold to beat to get a positive composable SKF. The

deviation of the simulation traces in (**a**) from the experimental data between 1 and 5 s is due to the slight increase in $\xi_u$. **c**, **d** Comparison of confidence intervals derived in this manuscript (Beta; solid-red trace and Gaussian; dotted-green trace) with those derived in the original composable security proof (ref. [10]; dashed-blue trace) as a function of $N$. Using the confidence intervals from ref. [10] leads to no key generation until almost the end (filled-blue square in (**a**) at $N/B \approx 10$).

attenuation factor of 0.2 dB/km). This should be achievable with some improvements in the hardware as well as the digital signal processing. We therefore expect that in the future, users across a point-to-point link could use the composable keys from our implementation to enable real applications such as secure data encryption, thus ushering in a new era for CVQKD.

## Methods

### Penalty from the asymptotic equipartition property

In ref. [25], the asymptotic equipartition property bound is proven in Corollary 6.5:

$$\frac{1}{n} H_{\min}^{\delta}(X^n | E^n) \geq H(X|E) - \frac{\Delta_{\text{AEP}}(\delta, \upsilon)}{\sqrt{n}}, \tag{8}$$

where

$$\Delta_{\text{AEP}}(\delta, \upsilon) := 4 \sqrt{\ell(\delta)} \log_2 \upsilon, \tag{9}$$

and

$$\upsilon \leq \sqrt{2^{-H_{\min}(X|E)}} + \sqrt{2^{H_{\max}(X|E)}} + 1, \tag{10}$$

$$\ell(\delta) := -\log_2 \left(1 - \sqrt{1 - \delta^2}\right). \tag{11}$$

In the following, we use the fact that $H_{\min}(X|E)$ is non-negative for our classical-quantum state, a proof of which is given in Supplementary Note 2.

$$H_{\min}(X|E) \geq 0 \Rightarrow 2^{-H_{\min}(X|E)} \leq 1, \tag{12}$$

$$H_{\max}(X|E) \leq \log_2 2^{2d} \Rightarrow \sqrt{2^{H_{\max}(X|E)}} \leq \sqrt{2^{2d}} = 2^d. \tag{13}$$

where $d$ denotes the number of bits per quadrature used during discretization.

Using the above relations in Eq. (10) allows us to bound $\upsilon$:

$$\upsilon \leq \sqrt{2^{-H_{\min}(X|E)}} + \sqrt{2^{H_{\max}(X|E)}} + 1 \leq 2^d + 2. \tag{14}$$

Now we can easily check that for $d > 1$,

$$\log(2^d + 2) < d + 1, \tag{15}$$

and that

$$\ell(\delta) < \log_2 \frac{2}{\delta^2}. \tag{16}$$

Putting all together we finally obtain

$$\Delta_{\text{AEP}}(\delta, d) \leq 4(d + 1) \sqrt{\log_2 \frac{2}{\delta^2}}. \tag{17}$$

### Penalty from entropy estimation

The entropy $H(\bar{Y})$ in Eq. (5) can be estimated from the empirical frequency

$$f(y_j) = \frac{n'(y_j)}{n'}, \tag{18}$$

where $n'(y_j)$ is the number of times a specific complex symbol $y_j = q_{\text{rx}}^j + i p_{\text{rx}}^j$ is obtained, and $n'$ is the total number of exchanged and corrected quantum symbols. One can define an entropy estimator

$$\hat{H}(\bar{Y}) = -\sum_j f(y_j) \log[f(y_j)]. \tag{19}$$

which is linked to $H(\bar{Y})$ by the following inequality[10,42]:

$$H(\bar{Y}) \geq \hat{H}(\bar{Y}) - \log(n') \sqrt{\frac{2 \log(2/\epsilon_{\text{ent}})}{n'}}. \tag{20}$$

This holds true up to a probability smaller than $\epsilon_{\text{ent}}$.

## Composable security against general attacks

For CVQKD with coherent states, the only known proofs providing composable security against general attacks[11,15] requires dual quadrature detection. This rules out the experiment in ref. 21, as despite recording the largest $N = 10^{11}$ symbols and the lowest $\xi_u$ value amongst all CVQKD works in Table 1, it used homodyning. On the upside, the proofs permit the assumption that the underlying quadrature data follows a Gaussian distribution, which somewhat relaxes the requirements on $N$. For instance, in the case of confidence intervals, one can observe the dotted-green traces in Fig. 3c and d show the best performance.

Nevertheless, to achieve composable security against general attacks, one needs $\epsilon^{(\text{gen})} \sim O(N^4)\epsilon^{(\text{coll})}$ as the final security parameter. A reasonable $\epsilon^{(\text{gen})}$ of $10^{-9}$ assuming $N \sim 10^8$ then requires $\epsilon^{(\text{coll})} < 10^{-41}$ but this is not the case with our current setup as $\epsilon^{(\text{coll})} \gtrsim \epsilon_{\text{qrng}} = 2 \times 10^{-6}$ actually. This limitation, due to the ADC digitization error in the QRNG, could be improved using longer measurement periods[30]. Yet another issue is the symmetrization requirement, a procedure in which Alice and Bob need to multiply their respective symbol trains by an identical random orthogonal matrix of size $N \times N$, which poses a major computational challenge.

## Reporting summary

Further information on research design is available in the Nature Research Reporting Summary linked to this article.

## Data availability

The data used in making some of the plots in Fig. 3 of the article have been deposited in the DTU database (https://doi.org/10.11583/DTU.20198891.v1). All other data are available from the corresponding authors upon reasonable request.

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

## Acknowledgements

We thank Marco Tomamichel for discussions regarding the security analysis. The work presented in this paper has been supported by the European Union's Horizon 2020 research and innovation programmes CiViQ (grant agreement no. 820466, concerned authors: N.J., H.M.C., H.M., D.S.N., A.K., S.P., B.O., C.P., T.G., and U.L.A.) and OPENQKD (grant agreement no. 857156, concerned authors: N.J., H.M.C., H.M., B.O., C.P., T.G., and U.L.A.). We also acknowledge support from the Innovation Fund Denmark (CryptQ, 0175-00018A, concerned authors: NJ, HMC, HM, TBP, TG, and ULA) and the Danish National Research Foundation (bigQ, DNRF142, concerned authors: N.J., H.M.C., H.M., D.S.N., T.G., and U.L.A.). C.L. and S.P. acknowledges funding from the EPSRC Quantum Communications Hub, Grant No. P/M013472/1 and EP/T001011/1.

## Author contributions

T.G. and U.L.A. conceived and supervised the experiment. N.J. designed the setup, conducted the experiments, and performed the final data analysis with help from T.G., H.M.C., A.K., and D.S.N. H.M.C. designed the digital signal processing framework. H.M. implemented the information reconciliation and privacy amplification with inputs from B.O., C.P., T.G., and T.B.P. C.L., M.K., and S.P. contributed to the security proof and provided theoretical support. N.J. and T.G. wrote the manuscript with contributions from all authors.

## Competing interests

The authors declare no competing interests
