## [Peer Review File · Nature Communications]

REVIEWER COMMENTS

Reviewer #1 (Remarks to the Author):

In the manuscript "Practical continuous-variable quantum key distribution with composable security", authors provide a theoretical improvement for the composable secret key rate calculation of Gaussian-modulated coherent state CVQKD protocol and present an experimental implementation of this protocol with technical improvements. Due to these improvements, their system is able to generate secret keys for a relatively small number of signals under the assumption of collective attacks.

There are several issues with the current manuscript in terms of presentation and significance of the work. A revision seems necessary.

1. In this work, authors consider only collective attacks without any discussion about the most general attacks – coherent attacks. Since the ultimate goal is to prove the composable security against the most general attacks, it is not sufficient to consider only collective attacks. Authors should at least discuss whether their method can provide any positive key rate for coherent attacks. It is likely that due to the limitation of available theoretical tools, a lifting from collective attacks to general attacks may result in zero key rate for the current implementation. If it is the case, authors should at least point out the limitation and leave it for a future theoretical improvement to overcome this limitation.

2. Authors should clearly discuss the novel contributions of this work, from both theoretical and experimental aspects. In particular, it is important to explain why their work is not just incremental improvement over the previous works. For the theoretical aspect, it seems this work follows closely the work by Leverrier (2015) (Ref. [4] of main text) with an improvement for the confidence interval estimation and an improvement of the correction term for asymptotic equipartition property. While authors mention that these improvements lead to an order of magnitude improvement for block size requirement to achieve the same key rate, it is probably desirable to have a comparison plot showing the secret key rate versus block size for these two approaches. For the experimental aspect, a discussion about other recent Gaussian-modulation CVQKD experimental demonstrations seems missing. For example, authors may consider discussing the differences between their experiment and the work in Phys. Rev. Lett. 125, 010502 (2020). Authors should discuss why their contributions from the experimental aspect are also not just incremental improvements. A key rate plot that puts different experiments as data points is also desirable.

3. At the end of Section II, authors remark on the technical limitation and mention that the impact of digitization is sufficiently minimized. Can authors explain whether their analysis uses Refs. [23, 24] to properly take into account the small gap in the secret key rate due to the digitization or they just assume a perfect Gaussian modulation for simplicity of analysis?

4. There are many minor presentation issues and typos to be fixed.

(a) In the main text, there are several unexplained symbols and abbreviations. Some of them are only explained in the supplementary material and some are explained much later in the text instead of the first appearance. For example, the invcdf function is not explained in the main text. The abbreviation RF is not introduced. Abbreviations in Fig. 1 are not introduced, and they are only explained in Fig. 2. The symbol \bar{Y} in Eq. (1) is also not explained.

(b) At the end of Introduction, authors say "we obtain > 53.4 Mbits worth of composable secure key material in the worst case." Can authors provide some contexts (conditions) for this statement of key rate? Under the parameters listed in Table 1 for 20 km fibers? Collective attacks?

(c) Authors use photon number units instead of shot noise unit. Their motivation is to compare with discrete-variable protocols. However, in this work, they do not have any direct comparison with discrete-variable protocols. On the other hand, it also makes a direct comparison with previous CV experiments more difficult. I would suggest presenting values in both units for both purposes instead of just PNU.

(d) In Fig. 3 (a), what is the meaning of negative key fraction?

(e) In Eq. (1) of main text (also in Eq. (62) of the supplementary material), the inequality sign seems to be flipped.

(f) In the abstract, authors claim "first" Gaussian-modulated coherent state CVQKD system that achieves certain performance. I feel a claim for first is not necessary. If a work does not contain any novel contribution, it does not worth publication. If one wants to claim first, one can always find a suitable condition for being first. This is meaningless.

(g) In the second sentence of the Introduction section, the language is not rigorous enough. There is no perfectly secure QKD protocol. The security definition of QKD is epsilon-security. With a very tiny probability, the QKD protocol may fail, that is, it does not abort but generates an insecure key such that when it is used to encrypt a message, Eve can actually break the confidentiality of the encrypted messages without being detected. I would suggest rephrasing the sentence in the context of composable security to make it more precise.

Reviewer #2 (Remarks to the Author):

The manuscript addresses very timely topic of finding practical solutions for quantum secure communication. Continuous variable communication has advanced a lot in the recent years however there are still some important hurdles to overcome. One of them is more challenging security proofs, so that in many cases only loose bounds can be given, limiting achievable distances and key rates. The manuscript responds to one of such challenges. It solves the problem of better account for the finite size effects in CVQKD and addresses the question of composability. The results are sound, relevant and will definitely have significant impact now, when more and more QKD systems are deployed under real world conditions. The new bounds derived from the presented security proofs allow for substantial improvements in distances and key rates. The theoretical work is supported by experiment results, which are very convincing. I can highly recommend manuscript for publication.

However there are few deficiencies in the presentation of results. Overall the paper is very well written, but in many parts very technical. Taking into account the wide scientific audience of Nature Communication, I would suggest adding more introductory and explanatory remarks. For example, introductory paragraph on composability (lines 40-44) would profit from more explanations. Also the idea of "non-composable proof" (line 48) will not be clear to less specialist reader.

Further, it is very commendable that the authors provide concrete figures of merit to characterize their system. However some reference to assess these results is not provided in sufficient detail and it is difficult to benchmark them. How the numbers achieved here compare to the state of art? The authors cite the corresponding literature, but would be good to have a direct comparison in the paper itself. Otherwise, I can only reiterate that the results are excellent and address important open problems, paper will be a valuable addition to Nature Communications and should be published after the comments above are addressed.

Reviewer #3 (Remarks to the Author):

QKD requires a large number of symbols N in order to guarantee a high level of confidence in the generated keys. In particular, when the key needs to satisfy composability conditions in CV-QKD, N rises above 10^9 to have practical secret key rates. The authors propose a new method that tightens the confidence intervals (based on properties of the beta function), improving the performance at lower values of N , using Gaussian modulation and satisfying composability.

The document covers the theoretical analysis of the tighter bounds and presents an experimental implementation of the new features in a proof-of-principle scenario that can be a good representative of 20 km fiber link, although it has important simplifications with respect to a possible commercial system.

As far as I understand it, the analysis in the article is correct and the topic is relevant for the CV-QKD community, since it partially alleviates the need to work with large blocks of data. I recommend its publication in Nature Communications, but I would suggest some improvements:

1. Although the main elements of a potential commercial setup are present in the experiment, it is far from a practical one, specially in terms of calibration and independence between Alice and Bob (e.g., sharing trigger and clock). I think it is necessary to indicate clearly in the main text that it is a proof-of-principle experiment, and mention the parts that would need to be provided in order to have a practical system.

1.a. The experimental results seem to be focussed on a one-shot measure, which is reasonable for clarity in my opinion, but it would be interesting to read an extrapolation to multiple measurements.

1.b. As it is mentioned in the supplementary, there is a clock and trigger connection between Alice and Bob. To be fair, I think it should be somehow reflected in the schemes.

1.c. I assume some of the components used in the experiment are homemade, but it would be interesting to indicate their main characteristics, and when possible the model (if it is commercial).

2. The document is clear in general, but not very homogeneous in style. In particular I found the introduction unbalanced. Maybe some additional references (e.g., to collective attacks in line 59) or the rephrasing of certain sentences would help give a better impression.

3. Some minor details in the main article:

3.a. Mention distance or attenuation in line 66.

3.b. The quality of FIG.1 is not very good in my pdf. If it is possible, a vector version would be nicer.

3.c. In line 111, as it is written, it can give the impression that the authors artificially detune the laser, which I understand is not the case, right?

4. Some minor details in the supplementary:

4.a. In line 24, where it says Gamma-distribution, shouldn't it be beta?

4.b. In line 112, what is the extension of the digital RRC filter (in samples)?

We would like to thank the reviewers for their positive reviews and valuable feedback. We truly appreciate their comments in order to make the manuscript better. We have addressed their concerns in this letter (our responses are highlighted in blue) and updated the manuscript accordingly. The complete list of changes is mentioned at the end of this letter.

Thanking you,

Sincerely,

Nitin Jain

(on behalf of all co-authors)

Reviewer 1

In the manuscript “Practical continuous-variable quantum key distribution with composable security”, authors provide a theoretical improvement for the composable secret key rate calculation of Gaussian-modulated coherent state CVQKD protocol and present an experimental implementation of this protocol with technical improvements. Due to these improvements, their system is able to generate secret keys for a relatively small number of signals under the assumption of collective attacks.

Response: We thank the reviewer for her/his concise summary of our work. In the following, we address the concerns raised by her/him.

1. In this work, authors consider only collective attacks without any discussion about the most general attacks – coherent attacks. Since the ultimate goal is to prove the composable security against the most general attacks, it is not sufficient to consider only collective attacks. Authors should at least discuss whether their method can provide any positive key rate for coherent attacks. It is likely that due to the limitation of available theoretical tools, a lifting from collective attacks to general attacks may result in zero key rate for the current implementation. If it is the case, authors should at least point out the limitation and leave it for a future theoretical improvement to overcome this limitation.

Response: We appreciate this comment from the reviewer. Per Refs. Leverrier (2017); Pirandola (2021), to achieve composable security against general attacks, one needs $\epsilon^{(\text{gen})} \sim O(N^4)\epsilon^{(\text{coll})}$ as the final security parameter, with N denoting the number of available quantum symbols. A reasonable $\epsilon^{(\text{gen})}$ of 10^{-9} assuming $N \sim 10^8$ then requires $\epsilon^{(\text{coll})} < 10^{-41}$ but this is not the case with our current setup as $\epsilon^{(\text{coll})} \gtrsim \epsilon_{\text{qrng}} = 2 \times 10^{-6}$ from our quantum random number generator (QRNG). Yet another issue is the symmetrization requirement, a procedure in which Alice and Bob need to multiply their respective symbol trains by an identical random orthogonal matrix of size $N \times N$, which poses a major computational challenge. We have now added these details in the paper.

2. Authors should clearly discuss the novel contributions of this work, from both theoretical and experimental aspects. In particular, it is important to explain why their work is not just incremental improvement over the previous works. For the theoretical aspect, it seems this work follows closely the work by Leverrier (2015) (Ref. [4] of main text) with an improvement for the confidence interval estimation and an improvement of the correction term for asymptotic equipartition property. While authors mention that these improvements lead to an order of magnitude improvement for block size requirement to achieve

the same key rate, it is probably desirable to have a comparison plot showing the secret key rate versus block size for these two approaches. For the experimental aspect, a discussion about other recent Gaussian-modulation CVQKD experimental demonstrations seems missing. For example, authors may consider discussing the differences between their experiment and the work in Phys. Rev. Lett. 125, 010502 (2020). Authors should discuss why their contributions from the experimental aspect are also not just incremental improvements. A key rate plot that puts different experiments as data points is also desirable.

Response: While we agree some kind of comparison is desirable, a key rate plot can be a rather dubious¹ way of comparing different works in our opinion. A secret key rate (or length or fraction) formula consists of several terms, and a comparison is only useful if several base parameters can be kept the same, so that the influence of one or two important terms can be understood. This is unfortunately not easy. For instance, the final numerical value shown in a graph showing some secret key output under finite-size corrections heavily depends on the block size, the epsilons used in the security analysis, etc., but these vary a lot from one work to the other. Also, not all publications actually do a proper information reconciliation: they just assume a certain efficiency and even neglect the frame error rate (FER) ! A re-calculation (to reset the base and make the comparison as fair as possible) in most cases is therefore a futile exercise, especially as the requisite details about the individual parts making up the sum are rarely disseminated in a proper manner.

What can be usefully compared are individual terms in the key length expression, which is what we indeed do via Figs. 3c and 3d, for instance. We have now also calculated the secret key fraction (SKF) using the confidence intervals from the 2015 work by Leverrier (now Ref. [7] in the paper) and found that it does not yield a composable key until $N = 10^9$, at which the worst-case SKF is 6.04×10^{-4} , which is almost two orders of magnitude lower than what we have achieved with our confidence intervals method. We have added a trace to Fig. 3 as well as added sentences in the figure caption and main text to summarize this result.

Furthermore, we have also added a table that provides a comparison of several ‘experimental’ aspects from 2 DV and 4 CVQKD implementations with similar physical channels, i.e., optical fiber spool with loss in the 4 – 5 dB regime, as ours. (The referee’s suggested citation, i.e., Phys. Rev. Lett. 125, 010502 (2020), is also one of the 4 CVQKD implementations.) The parameters include mean photon number of Alice’s state alphabet, operating repetition rate / quantum data bandwidth of the QKD system, obtainable secret key fraction, etc.

3. At the end of Section II, authors remark on the technical limitation and mention that the impact of digitization is sufficiently minimized. Can authors explain whether their analysis uses Refs. [23, 24] to properly take into account the small gap in the secret key rate due to the digitization or they just assume a perfect Gaussian modulation for simplicity of analysis?

Response: We assume perfect Gaussian modulation for simplicity, and have added this sentence as well.

4. There are many minor presentation issues and typos to be fixed.
 - (a) In the main text, there are several unexplained symbols and abbreviations. Some of them are only explained in the supplementary material and some are explained much later in the text instead of the first appearance. For example, the invcdf function is not explained in the main text. The abbreviation RF is not introduced. Abbreviations in Fig. 1 are not introduced, and they are only explained in Fig. 2. The symbol \bar{Y} in Eq. (1) is also not explained.

Response: We thank the referee for this thorough scrutiny. We have now updated the text and figures to take these points into account. In particular, we have revamped the entire introduction,

¹Consider two QKD systems E1 and E2 that produce keys at a rate $r = 10$ kbps after finite-size corrections. The system repetition rate in E1 is 10 GHz while in E2 is 1 MHz. Based on only r , both systems are equally good, whereas in reality, a key fraction of 10^{-6} implies E1 is probably much less robust than E2, i.e., might stop producing keys with just a bit of fluctuations.

and revised Fig. 1 in a way that eliminates the requirement of having to explain the abbreviations.

- (b) At the end of Introduction, authors say “we obtain > 53.4 Mbits worth of composable secure key material in the worst case.” Can authors provide some contexts (conditions) for this statement of key rate? Under the parameters listed in Table 1 for 20 km fibers? Collective attacks?

Response: We have now provided the necessary context. The paragraph now reads:

After taking finite-size effects as well as confidence intervals from various system calibrations into account, we achieve a positive composable key length with merely $N \approx 2 \times 10^8$ coherent states (also referred to as ‘quantum symbols’ from hereon) transmitted over a 20 km long fiber-optic channel. With $N = 10^9$, we obtain > 41 Mbits worth of key material that is composable secure against collective attacks, assuming worst-case confidence intervals.

- (c) Authors use photon number units instead of shot noise unit. Their motivation is to compare with discrete-variable protocols. However, in this work, they do not have any direct comparison with discrete-variable protocols. On the other hand, it also makes a direct comparison with previous CV experiments more difficult. I would suggest presenting values in both units for both purposes instead of just PNU.

Response: We have now added a table that compares parameters such as the deployed mean photon number of Alice’s quantum state alphabet in both CV and DV experiments. We feel presenting values in both units would make the manuscript rather cluttered. Instead, we have now mentioned at multiple locations that SNU and PNU values have just a factor of 2 between them. Finally, we also cite the work by Lasota et al. (2017) which analyses DVQKD protocols by using the excess channel noise model—typically used in CVQKD protocols—to compare the performance and robustness of these two families. Notably, the main parameter that facilitates this comparison is the mean number of noise photons.

- (d) In Fig. 3 (a), what is the meaning of negative key fraction?

Response: We have now updated the graphs to show only strictly positive values of the key fraction in both the experimental data and simulation.

- (e) In Eq. (1) of main text (also in Eq. (62) of the supplementary material), the inequality sign seems to be flipped.

Response: The equation strives to state that the sum of the terms on the RHS is a **lower bound** on the number of secret bits that can be extracted from the raw key.

- (f) In the abstract, authors claim “first” Gaussian-modulated coherent state CVQKD system that achieves certain performance. I feel a claim for first is not necessary. If a work does not contain any novel contribution, it does not worth publication. If one wants to claim first, one can always find a suitable condition for being first. This is meaningless.

Response: We have modified the claim to remove “first” from the sentence.

- (g) In the second sentence of the Introduction section, the language is not rigorous enough. There is no perfectly secure QKD protocol. The security definition of QKD is epsilon-security. With a very tiny probability, the QKD protocol may fail, that is, it does not abort but generates an insecure key such that when it is used to encrypt a message, Eve can actually break the confidentiality of the encrypted messages without being detected. I would suggest rephrasing the sentence in the context of composable security to make it more precise.

Response: The reviewer is completely right. However, to prevent the introduction from becoming detailed/convoluted, we have now added *In an ideal case*, in the beginning of the sentence.

Reviewer 2

The manuscript addresses very timely topic of finding practical solutions for quantum secure communication. Continuous variable communication has advanced a lot in the recent years however there are still some important hurdles to overcome. One of them is more challenging security proofs, so that in many cases only loose bounds can be given, limiting achievable distances and key rates. The manuscript responds to one of such challenges. It solves the problem of better account for the finite size effects in CVQKD and addresses the question of composability. The results are sound, relevant and will definitely have significant impact now, when more and more QKD systems are deployed under real world conditions. The new bounds derived from the presented security proofs allow for substantial improvements in distances and key rates. The theoretical work is supported by experiment results, which are very convincing. I can highly recommend manuscript for publication.

Response: We thank the reviewer for his/her nice recommendation of our article. We hope that in the following we can address the concerns raised by him/her.

However there are few deficiencies in the presentation of results. Overall the paper is very well written, but in many parts very technical. Taking into account the wide scientific audience of Nature Communication, I would suggest adding more introductory and explanatory remarks. For example, introductory paragraph on composability (lines 40-44) would profit from more explanations. Also the idea of "non-composable proof" (line 48) will not be clear to less specialist reader. Further, it is very commendable that the authors provide concrete figures of merit to characterize their system. However some reference to assess these results is not provided in sufficient detail and it is difficult to benchmark them. How the numbers achieved here compare to the state of art? The authors cite the corresponding literature, but would be good to have a direct comparison in the paper itself. Otherwise, I can only reiterate that the results are excellent and address important open problems, paper will be a valuable addition to Nature Communications and should be published after the comments above are addressed.

Response: We thank the reviewer for these critical yet constructive remarks, which we believe are now addressed in the latest version of our manuscript. To elaborate, we have revamped the entire introduction to make it more general and less technical and the term "non-composable" has now been removed. Lastly we have added a table that provides a comparison of several parameters from 2 DV and 4 CV experiments conducted over similar physical channels (fiber-optic implementations with loss in the 4 – 5 dB regime). The parameters include mean photon number of Alice's state alphabet, operating repetition rate / quantum data bandwidth of the QKD system, obtainable secret key fraction, etc.

Reviewer 3

QKD requires a large number of symbols N in order to guarantee a high level of confidence in the generated keys. In particular, when the key needs to satisfy composability conditions in CV-QKD, N rises above 10^9 to have practical secret key rates. The authors propose a new method that tightens the confidence intervals (based on properties of the beta function), improving the performance at lower values of N , using Gaussian modulation and satisfying composability.

The document covers the theoretical analysis of the tighter bounds and presents an experimental implementation of the new features in a proof-of-principle scenario that can be a good representative of 20 km fiber link, although it has important simplifications with respect to a possible commercial system.

As far as I understand it, the analysis in the article is correct and the topic is relevant for the CV-QKD community, since it partially alleviates the need to work with large blocks of data. I recommend its publication in Nature Communications, but I would suggest some improvements:

Response: We thank the reviewer for the nice summary of our work and his/her recommendation and hope to address the raised concerns in the following.

1. Although the main elements of a potential commercial setup are present in the experiment, it is far from a practical one, specially in terms of calibration and independence between Alice and Bob (e.g., sharing

trigger and clock). I think it is necessary to indicate clearly in the main text that it is a proof-of-principle experiment, and mention the parts that would need to be provided in order to have a practical system.

- (a) The experimental results seem to be focussed on a one-shot measure, which is reasonable for clarity in my opinion, but it would be interesting to read an extrapolation to multiple measurements.
- (b) As it is mentioned in the supplementary, there is a clock and trigger connection between Alice and Bob. To be fair, I think it should be somehow reflected in the schemes.
- (c) I assume some of the components used in the experiment are homemade, but it would be interesting to indicate their main characteristics, and when possible the model (if it is commercial).

Response: We have now updated Fig. 2 to show clock sharing and triggering explicitly, apart from also adding a sentence in the main text. We also mention at multiple locations now that our experiment is proof-of-concept in nature. We now cite Gehring et al. (2021) with regards to our homemade (heterodyne) detector, and have added vendor names for various devices used in the setup. Finally, we have multiple (shorter) measurements that confirm the basic stability of the setup.

- 2. The document is clear in general, but not very homogeneous in style. In particular I found the introduction unbalanced. Maybe some additional references (e.g., to collective attacks in line 59) or the rephrasing of certain sentences would help give a better impression.

Response: We have now revised the entire introduction, added a new version of Fig. 1 and a new table that provides a comparison of several parameters from 2 DV and 4 CV experiments conducted over similar physical channels.

- 3. Some minor details in the main article:

- (a) Mention distance or attenuation in line 66.
- (b) The quality of FIG.1 is not very good in my pdf. If it is possible, a vector version would be nicer.
- (c) In line 111, as it is written, it can give the impression that the authors artificially detune the laser, which I understand is not the case, right?

Response: We have added information (in the vicinity of line 66) about the distance as well as that the keys are obtained against the assumption of collective attacks. Also, we use a vector graphics version of Fig. 1 now. Finally, we do indeed—before a QKD measurement—vary the center frequency of the Rx laser (by means of temperature tuning) so that the beat between the Tx and Rx lasers is around 320 MHz.

- 4. Some minor details in the supplementary:

- (a) In line 24, where it says Gamma-distribution, shouldn't it be beta?

Response: Per <https://mathworld.wolfram.com/Chi-SquaredDistribution.html> it would seem that Gamma distribution is correct.

- (b) In line 112, what is the extension of the digital RRC filter (in samples)?

Response: The RRC filter had a span of 20 symbols. We have put this in the Supplement.

List of changes

Note that we made some improvements in our discretization strategy, fixed a minor mistake in our code for secret key length calculation (namely, included a pre-factor due to the non-zero frame error rate), and performed information reconciliation (IR) with a different and better error correcting code. The final result is

that while the threshold at which we start getting a positive secret key length came down from $N \lesssim 3.5 \times 10^8$ to $N \approx 2 \times 10^8$ coherent states, the overall secret key fraction is slightly lower, and so, at $N = 10^9$, we obtain 41.4 Mbits (instead of 53.4 Mbits) worth of key material. We have updated the manuscript with these new values and the Supplement with the details of the changed IR procedure. Furthermore, Table 1 has now been completely revised: it now presents a comparison among different CV and DVQKD experiments from the last decade with similar channel loss/distance. Lastly, we have also re-structured the manuscript to have it in a format closer to what the journal expects for publication.

Below is a section-wise list specifying details of all the *significant* changes made to the manuscript.

***Introduction* (Pages 1-3)**

1. Complete revamp of Fig. 1. (changed to a vector graphics version, added an encryption channel, removed the transmitter and receiver boxes) and updated caption.
2. Elaborated terms such as amplitude and phase quadratures for generalization.
3. Consolidated the important elements of security proof (type of attacks, finite size regime, compositability) in a single paragraph and improved the overall explanation.

***Composably secure key* (Pages 3-4)**

1. Added explanations and elaborated several terms.
2. Removed the acronym GMCS.

***Experiment* (Pages 4-6)**

1. Updated Fig. 2 to show clock sharing and triggering explicitly, apart from also adding a sentence in the main text.
2. Added model/vendor name for the devices used in the transmitter and receiver.
3. Added numerical information (from the erstwhile Table 1) in the main text under ‘Protocol operation’ subsection.

***Discussion* (Pages 6-8)**

1. Added a table that provides a comparison of several experimental parameters from 2 DV and 4 CVQKD implementations (including ours) with similar physical channels.
2. Updated various numerical values. Also added the secret key fraction obtained with the confidence intervals from Ref. 7 in the paper.
3. Added discussion about composable security against general attacks, as well as the limitations with our CVQKD system that prevent us from achieving it.
4. Updated Fig. 3(a) and 3(b) due to re-calculation of the secret key length.

***Bibliography* (Pages 8-9)**

The order of references in the bibliography has changed. Also, we have several new citations:

- A. Leverrier. Security of continuous-variable quantum key distribution via a gaussian de finetti reduction. *Phys. Rev. Lett.*, 118:200501, 2017.
- F. Xu *et al.* Experimental quantum key distribution with source flaws, *Phys. Rev. A* 92:032305, 2015.

- N. T. Islam *et al.* Provably secure and high-rate quantum key distribution with time-bin qudits, *Science Advances*, 3(11):e1701491, 2017.
- Y. Zhang *et al.* Long-Distance Continuous-Variable Quantum Key Distribution over 202.81 km of Fiber, *Physical Review Letters*, 125(1):10502, 2020.
- N. Jain *et al.* qTReX : A semi-autonomous continuous-variable quantum key distribution system. In *The Optical Fiber Communication Conference (OFC)*, Optica Technical Digest, pp. M3Z.2. Optica Publishing Group, 2022.
- M. Lasota, R. Filip, & V. C. Usenko. Robustness of quantum key distribution with discrete and continuous variables to channel noise. *Physical Review A*, 95(6), 1–13 (2017).

References

- Gehring, T., Lupo, C., Kordts, A., Solar Nikolic, D., Jain, N., Rydberg, T., Pedersen, T. B., Pirandola, S., and Andersen, U. L. (2021). Homodyne-based quantum random number generator at 2.9 Gbps secure against quantum side-information. *Nature Communications*, 12(1):1–11.
- Lasota, M., Filip, R., and Usenko, V. C. (2017). Robustness of quantum key distribution with discrete and continuous variables to channel noise. *Physical Review A*, 95(6):1–13.
- Leverrier, A. (2017). Security of continuous-variable quantum key distribution via a gaussian de finetti reduction. *Phys. Rev. Lett.*, 118:200501.
- Pirandola, S. (2021). Limits and security of free-space quantum communications. *Physical Review Research*, 3(1):013279.

REVIEWERS' COMMENTS

Reviewer #1 (Remarks to the Author):

I am satisfied with authors' responses to all previous comments. I understand the difficulty to compare different experimental works in a key rate plot and feel Table I is very helpful for this purpose. I am also glad that authors now directly compare their method with Ref. [7] in Fig. 3. One small minor issue with Fig. 3 (a) is that authors may try to find a better location to put the legend so that no data point is blocked. Authors said at the end of Fig. 3 caption that no key is generated (blue square in (a)) until at N is roughly 10^9 . But the data point at $N = 10^9$ is probably blocked by the legend. On the other hand, the data point at $N=10^{10}$ is almost zero. Is it probably a typo in the caption, that is, saying $N=10^{10}$ instead? Anyway, I am satisfied with the revision. I feel it is now in a good quality for publication.

Reviewer #2 (Remarks to the Author):

The authors have very carefully addressed all the points raised by reviewers. I do not agree that the way they have introduced continuous variable approach in the second paragraph of the introduction is the optimal one, but overall I am very satisfied with the revision of the manuscript. A commendable effort have been made in clarifying a number of important points mentioned in referee reports. I recommend the acceptance of the revised manuscript.

Reviewer #3 (Remarks to the Author):

The authors have notably upgraded the quality of the document with their revision. I recommend the publication of the article in Nature Communications.